

# Air transportation, population density and temperature predict the spread of COVID-19 in Brazil

Pedro Pequeno[1], Bruna Mendel[2], Clarissa Rosa[3], Mariane Bosholn[1], Jorge Luiz Souza[4], Fabricio Baccaro[5], Reinaldo Barbosa[1] and William Magnusson[3]

[1] Instituto Nacional de Pesquisas da Amazônia, Boa Vista, Brazil
[2] Universidade Federal de Roraima, Boa Vista, Brazil
[3] Instituto Nacional de Pesquisas da Amazônia, Manaus, Brazil
[4] Instituto Nacional da Mata Atlântica, Santa Teresa, Brazil
[5] Universidade Federal do Amazonas, Manaus, Brazil

## ABSTRACT

There is evidence that COVID-19, the disease caused by the betacoronavirus SARS-CoV-2, is sensitive to environmental conditions. However, such conditions often correlate with demographic and socioeconomic factors at larger spatial extents, which could confound this inference. We evaluated the effect of meteorological conditions (temperature, solar radiation, air humidity and precipitation) on 292 daily records of cumulative number of confirmed COVID-19 cases across the 27 Brazilian capital cities during the 1st month of the outbreak, while controlling for an indicator of the number of tests, the number of arriving flights, population density, proportion of elderly people and average income. Apart from increasing with time, the number of confirmed cases was mainly related to the number of arriving flights and population density, increasing with both factors. However, after accounting for these effects, the disease was shown to be temperature sensitive: there were more cases in colder cities and days, and cases accumulated faster at lower temperatures. Our best estimate indicates that a 1 °C increase in temperature has been associated with a decrease in confirmed cases of 8%. The quality of the data and unknowns limit the analysis, but the study reveals an urgent need to understand more about the environmental sensitivity of the disease to predict demands on health services in different regions and seasons.

## INTRODUCTION

The disease COVID-19, caused by the betacoronavirus SARS-CoV-2, has caused panic throughout the world by overwhelming medical services in many countries, leading to deaths that might have been avoided if patients had access to intensive-care units (ICUs). This has led to an unprecedented collaboration within and among countries to slow the spread of the disease, principally using social distancing (*Ebrahim et al., 2020*; *Wilder-Smith & Freedman, 2020*). While it is not clear how much present policies will reduce overall infection rates by SARS-CoV-2, there is consensus that slowing the spread of the

Corresponding author
Clarissa Rosa,
alvesrosa_c@hotmail.com

disease will save lives by tailoring patient demands to the capacity of health systems (*Ferguson et al., 2020*; *Walker et al., 2020*).

The strategies of social isolation applied in countries on all continents have allowed time for authorities to undertake interventions to strengthen their health systems, and one of the main actions is to estimate the number of cases of COVID-19 in each region (*Anderson et al., 2020*; *Walker et al., 2020*). This information is essential to scale the number of ICUs to the number of critically ill patients who normally require supportive lung ventilation (*Xu et al., 2020*). Brazil has a large per capita number of ICUs in comparison with Europe, but those units are not evenly spread among regions, with more ICUs per capita in southern states than in northern regions, leaving many Brazilians at large distances from the nearest ICU (*Rhodes & Moreno, 2012*). Moreover, the large demographic and socioeconomic discrepancies in the country create significant variation in susceptibility to infectious diseases (*Barreto et al., 2011*).

One of the problems in predicting the demand for hospital services is that the disease is new so that its behavior is still poorly understood, and the virus may be evolving rapidly (*Zhao et al., 2004*; *Yang et al., 2020*; *Morais et al., 2020*). Therefore, models developed in one country may give poor predictions in another. Habitat-specificity modeling suggests that SARS-CoV-2 spread may be related to environmental conditions, especially temperature and humidity (*Sajadi et al., 2020*; *Wang et al., 2020*). Further, at the host level, there is circumstantial evidence that COVID-19 is related to shortage of vitamin D, which could result from limited exposition to solar radiation (*Grant et al., 2020*). Indeed, it has been suggested that solar radiation might deactivate the virus (*Poole, 2020*). Although preliminary, these results provide a plethora of mechanistic processes linking weather and virus spread that need to be better understood.

Brazil is one of the largest countries in the world, spanning both hemispheres, with latitudes varying from 5°N to 33°S. This means that climatic conditions vary greatly and simple models that do not take into account the possible environmental sensitivity of COVID-19 might not be adequate to predict when and where there will be the greatest demand for health services in Brazil (Fig. 1). One difficulty in quantifying this sensitivity is that climate is likely to correlate with demographic and socioeconomic factors across larger spatial extents. Thus, environmental effects could be confounded unless risk factors for viral spread are taken into account, such as population density, transport connectivity and economic status (*Poole, 2020*; *Wang et al., 2020*; *Ribeiro et al., 2020*).

In an attempt to determine whether environmental variables have significant effects on the propagation of COVID-19, we modeled the daily cumulative number of confirmed cases among Brazilian capital cities in relation to meteorological variables during the 1st month of the disease in the country, while controlling for several demographic and socioeconomic factors. We used only capital cities because they are presently the only reliable sources of COVID-19 cases and represent much of the climatic variation within Brazil. Data on connectivity and frequency of cases is not presently adequate to model the spread of the disease at the municipal level, but this should be available in the future and can be used to test our hypotheses.

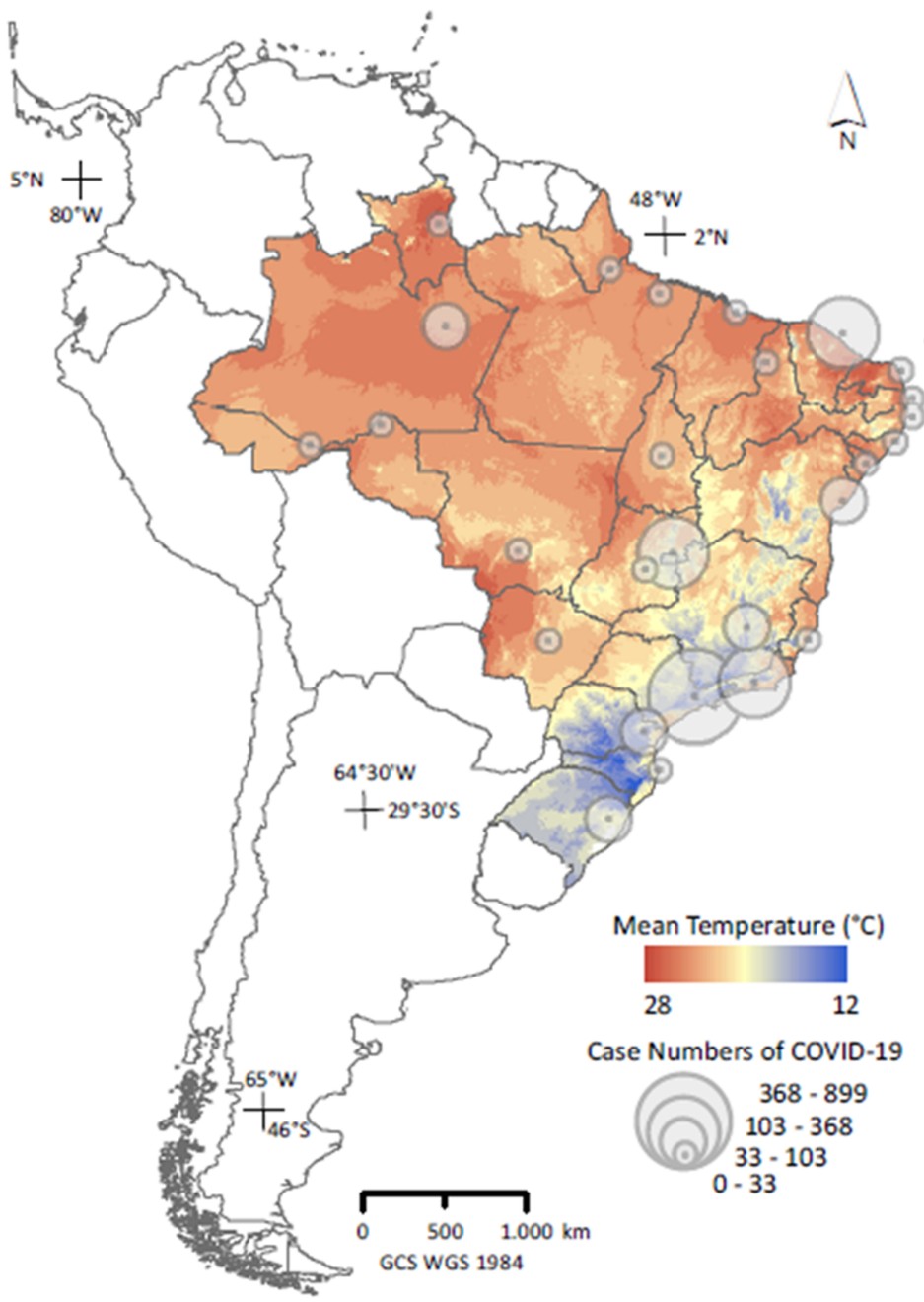

**Figure 1** **Distribution of counts of confirmed COVID-19 cases across capital cities in Brazil on 26 March, 2020 (*n* = 27); (*Secretarias de Saúde das Unidades Federativas, 2020*), superimposed on the country's thermal variability.** Temperature data represent means for March over 1989–2019 (*Fick & Hijmans, 2017*), and were used only in the map; actual analyses used current, daily meteorological data.                                               

## MATERIALS AND METHODS

We obtained daily cumulative counts of confirmed cases of COVID-19 for each of the 27 Brazilian state capital cities, as reported by State Health Secretaries and compiled by volunteers (Table S1; *Secretarias de Saúde das Unidades Federativas, 2020*).

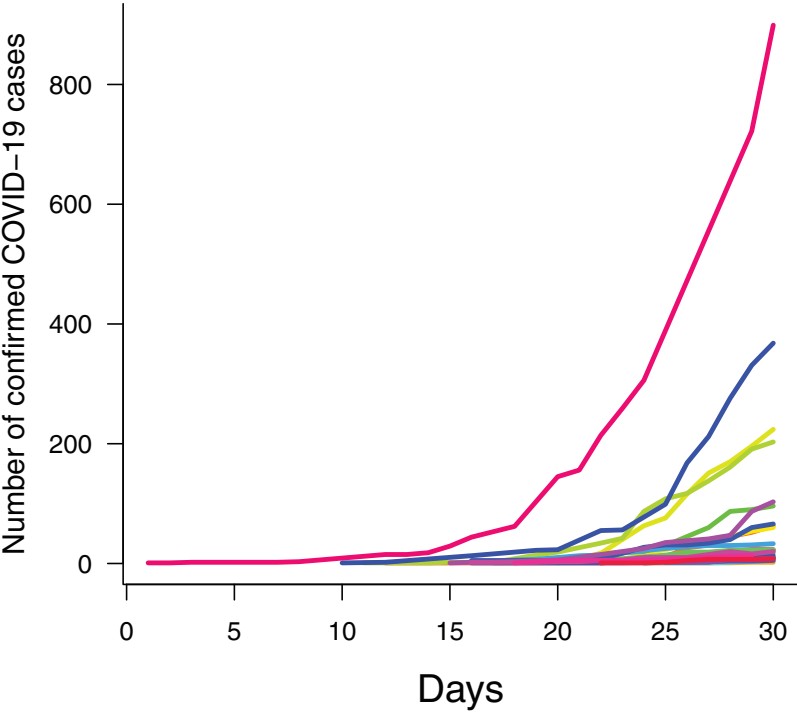

**Figure 2 Daily counts of confirmed cases of COVID-2019 across Brazilian state capital cities during the first month.** Exponential increase in daily counts of confirmed cases of COVID-2019 across Brazilian state capital cities ($n$ = 292) during the first monitored month. Each line/colour represents one capital city ($n$ = 27).

We focused on the month since the first confirmed case, from February 26 to March 26, 2020, for which there were 292 reports on daily counts across cities.

We considered several potential predictors of the number of confirmed cases. First, it was important to account for the number of tests for COVID-19, as performing more tests tends to reveal more positive cases (*Roser et al., 2020*). The Brazilian Government has not been systematically reporting the number of performed tests, but has recommended testing of all suspected patients with severe symptoms, and the Ministry of Health reported the number of suspected cases per state until March 18, 2020. Therefore, we used the number of suspected cases per state on that date as a proxy for the number of tests, under the reasonable assumption that states with more suspected cases performed more tests.

Further, we considered the following predictors: (1) time in days, to account for the exponential growth in case numbers during this period (Fig. 2); (2) number of arriving flights in the city's metropolitan area in 2020, as airline connections can facilitate the spread of the virus (*Ribeiro et al., 2020*); (3) city population density, to account for facilitation of transmission under higher densities (*Poole, 2020*); (4) proportion of elderly people (≥60 years old) in the population, assuming that the elderly may be more likely to show severe symptoms of SARS-CoV-2 and, thus, to be diagnosed with COVID-19; (5) citizen mean income, which may affect the likelihood of people being infected by the virus, for example, due to limited access to basic sanitation or limited social isolation capabilities; (6) and the following meteorological variables: mean daily temperature (°C),

mean daily solar radiation (kJ/m$^2$), mean daily relative humidity (%) and mean daily precipitation (mm). The number of suspected cases and socioeconomic variables only varied across cities, whereas meteorological variables varied both between and within cities.

Data on population density, the elderly and income were obtained for the last quarter of 2019 from the Brazilian Institute for Geography and Statistics (IBGE), which samples Brazilian households quarterly for socioeconomic indicators (*Sidra, 2020*) (Table S2). Flight data were obtained from the current statistical annuary of the Brazilian Agency for Civilian Aviation (ANAC) (Table S3; *ANAC, 2020*). Hourly meteorological data were obtained from the automatic stations maintained by the Brazilian Institute for Meteorology (INMET) in the capital cities (Table S4; *INMET, 2020*).

We investigated the response of case counts to putative predictors using a Generalized Linear Mixed Model (GLMM) assuming Poisson-distributed errors and log link, and using capital-city identity as a random factor to account for autocorrelated errors within cities. This formulation induces a compound symmetry correlation structure on residuals within cities, which is mathematically equivalent to that of classical, "repeated measures" linear models (*Zuur et al., 2009*). The numbers of suspected cases and arriving flights were log-transformed to account for their highly skewed distributions, and all predictors were scaled to zero mean and unit standard deviation to facilitate parameter estimation. Consequently, estimated coefficients were scaled, thus providing a measure of predictor relative importance. Specifically, assuming a log link, a change of one unit in the scaled predictor implies a mean percent change of (exp(coefficient/SD) − 1) × 100 in the number of confirmed cases, where "coefficient" and "SD" are the predictor's model coefficient and standard deviation, respectively.

We considered time lags in the effect of meteorological conditions. Incubation time of COVID-19 averages 5 days (*Lauer et al., 2020*), and case confirmation in Brazil has taken from several days to 2 weeks due to overload of test laboratories. Therefore, the time between infection and case confirmation is likely to be longer than a week. Accordingly, we considered a set of models including all predictors but varying in the number of days meteorological predictors were lagged relative to case counts, ranging from 7 to 30 days with daily steps. Then, models were compared with Akaike's Information Criterion (AIC), a standard measure of model relative support, and the model with the lowest AIC was judged as the most supported.

Precipitation time series had missing intervals for some capital cities. Therefore, we performed two versions of the above analysis: one including all predictors but excluding days for which precipitation was lacking ($n = 269$) and another one excluding precipitation as predictor and using all counts of confirmed COVID-19 cases ($n = 292$). Because both analyses produced largely similar results, with precipitation having a negligible model coefficient (Figs. S1 and S2; Table S5), we present the analysis using the larger sample.

Lastly, we considered possible interactions between time and other predictors, assuming some factors could accelerate the temporal increase in number of confirmed cases. By definition, GLMMs assuming a log link implicitly account for interactive effects to some
degree, as log-linear models imply multiplicative effects. Still, we ran a separate GLMM which explicitly included interaction terms between time and the remaining predictors. To avoid model overparameterization, we only used the significant predictors identified in the previous analysis.

For all models, we computed the conditional predictive power ($R_c^2$), which indicates the variance explained jointly by predictors and the random factor, and the marginal predictive power ($R_m^2$), which only considers predictor effects. In these calculations, only significant predictors were retained in the model to avoid inflation of explained variance due to spurious parameters.

We acknowledge that it would be better to directly model the spread of SARS-CoV-2, but we cannot do that without making assumptions about the relationship between infection by the virus and the appearance of symptoms of the disease, which may be related to the factors that we are investigating. All analyses were performed in R 3.6.3 (*R Core Team, 2020*), with aid of packages "coronabr" (*Mortara, Sánchez-Tapia & Martins, 2020*) and "covid19br" (*Paterno, 2020*) for assessing counts of suspected cases by state, "lme4" for GLMM (*Bates et al., 2015*), "MuMIn" for AIC and $R^2$ calculations (*Bartoń, 2019*), and "visreg" for visualization of predictor effects (*Breheny & Burchett, 2017*).

## RESULTS

There was strong support for the model whose meteorological predictors were lagged by 15 days, as indicated by its much lower AIC (Fig. 3). According to this model, the only significant predictors of the number of confirmed COVID-19 cases were time, the number of arriving flights, population density and temperature (Table 1). The number of confirmed cases increased with time (Fig. 4A), the number of arriving flights (Fig. 4B) and population density (Fig. 4C), whereas it decreased with temperature (Fig. 4D). Considering that model coefficients were scaled, comparing their values gives an indication of the relative importance of each predictor. Accordingly, time, the number of arriving flights and population density had the strongest effects (i.e., largest coefficients), followed by temperature (Table 1). Nevertheless, a change in 1 °C predicted a decrease in the number of confirmed cases by $(\exp(-0.26/3.11) - 1) \times 100 = 8\%$, independently of other factors (Table 1). Significant predictors explained 77% of the variance of daily counts of confirmed COVID-19 across capital cities in Brazil.

Explicitly accounting for interactions between time and the remaining predictors identified in the previous analysis suggested significant interactions between time and the number of arriving flights, and time and temperature, although the magnitude of the interaction coefficients was low (Table 2). On average, the temporal increase in confirmed COVID-19 cases began earlier in cities with more flights, causing a leftward shift in the relationship between confirmed cases and time (Fig. 5A). In parallel, the number of confirmed cases increased faster at lower temperatures, causing a steeper slope in the relationship between confirmed cases and time (Fig. 5B). However, these effects were relatively weak, and there was no improvement in predictive power (Table 2). Thus, the previous, simpler model captured the main patterns in the data very well.

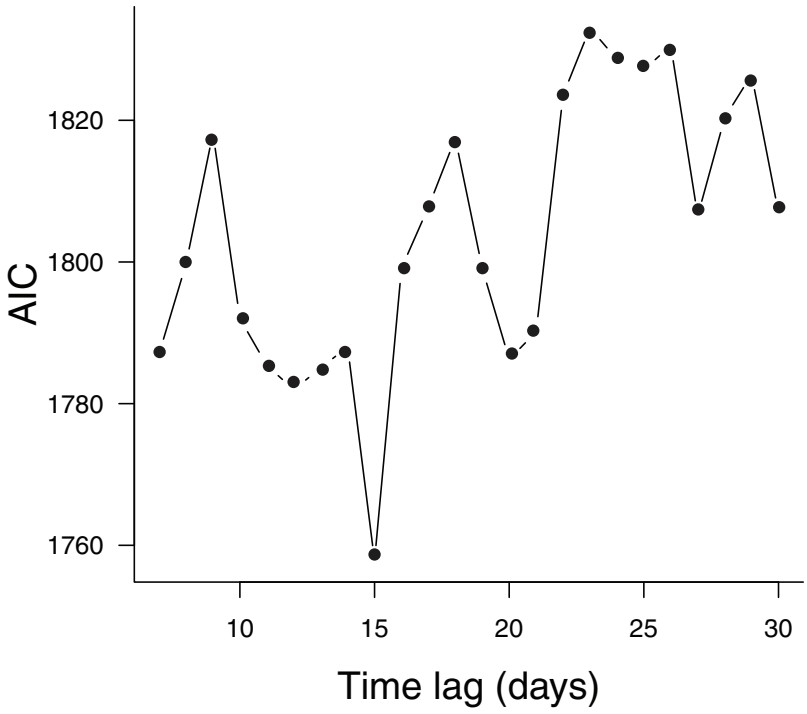

**Figure 3 Models considering different time lags in meteorological predictors.** Comparison of models considering different time lags in meteorological predictors using Akaike's Information Criterion (AIC). Each point represents one alternative model.

**Table 1 Results of the most supported Generalized Linear Mixed Model (GLMM) testing for independent effects on daily cumulative counts of confirmed COVID-19 across the 27 capital cities in Brazil ($n = 292$; $R_c^2 = 0.98$; $R_m^2 = 0.77$).** The model assumed Poisson-distributed errors and log link, and used capital city identity as a random factor to account for autocorrelated errors of time series within cities. All predictors were scaled to zero mean and unit standard deviation. SD indicates predictor standard deviation; numbers in bold represent statistically significant effects ($P < 0.05$). Variables were as follows: time—time elapsed in days; Log suspected—log-transformed number of suspected COVID-19 cases in March 18, 2020; Log flights—log-transformed number of arriving flights in 2020; density—inhabitants by $km^2$; elderly—proportion of elderly people ($\geq$ 60 years old); income—mean citizen income (R\$); temperature—mean daily temperature (°C) with a 15-day lag; radiation—mean daily solar radiation ($kJ/m^2$) with a 15-day lag; humidity—mean daily air humidity (%) with a 15-day lag.

| Predictor | SD | Coefficient | $z$ | $P$ |
|---|---|---|---|---|
| Intercept | – | 2.17 | – | – |
| Time | 5.72 | 1.4 | 67.7 | **<0.001** |
| Log suspected | 1,399.23 | 0.18 | 0.59 | 0.556 |
| Log flights | 351.06 | 0.86 | 3.2 | **0.001** |
| Density | 3,087.94 | 0.47 | 2.01 | **0.044** |
| Elderly | 3.6 | −0.36 | −1.59 | 0.111 |
| Income | 787.39 | 0.17 | 0.81 | 0.415 |
| Temperature | 3.11 | −0.26 | −5.87 | **<0.001** |
| Radiation | 321.67 | 0.04 | 1.52 | 0.127 |
| Humidity | 10.26 | 0.07 | 1.74 | 0.082 |

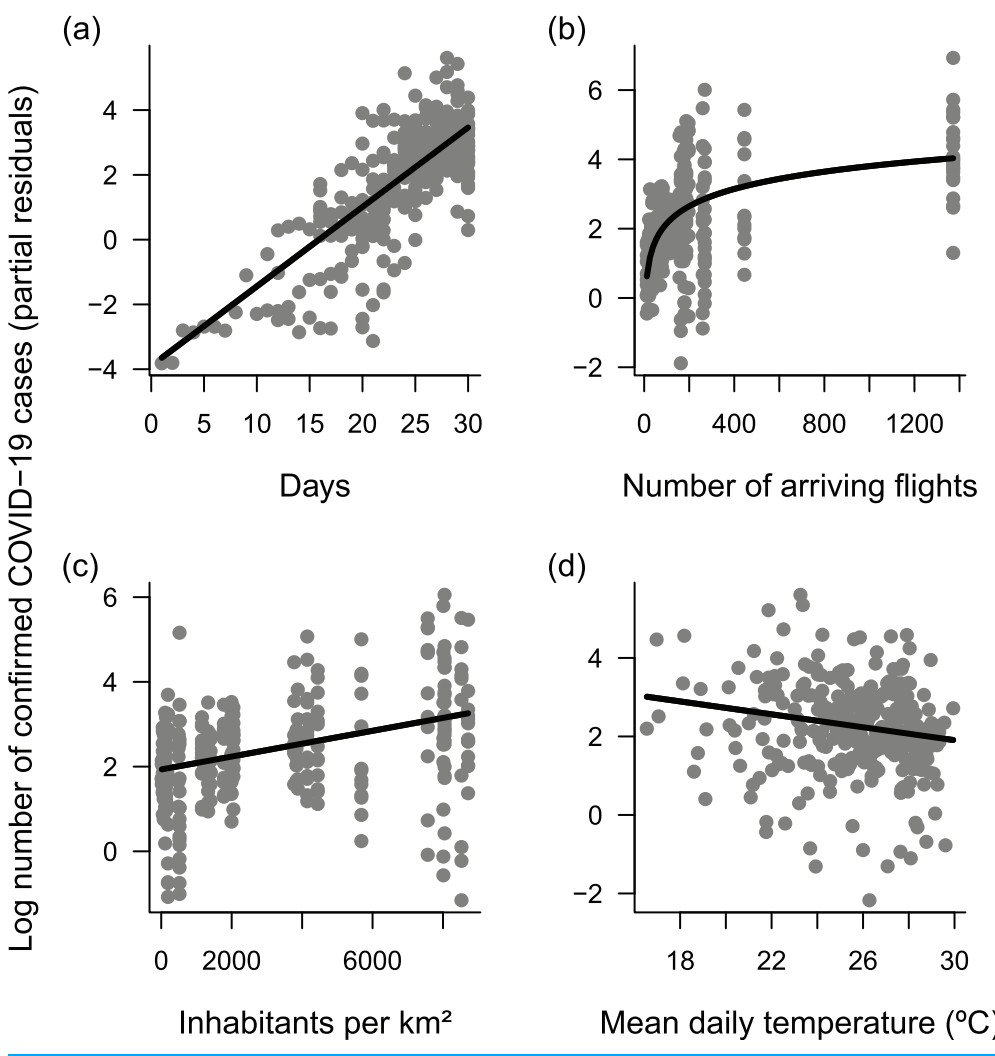

**Figure 4 Response of daily cumulative counts of confirmed COVID-19 cases across the 27 Brazilian capital cities.** (A) time, (B) number of arriving flights, (C) population density and (D) temperature, as indicated by the most supported Generalized Linear Mixed Model (GLMM). The model assumed Poisson-distributed errors and log link, and included capital city identity as a random factor to account for autocorrelated errors in time series within cities. Each point represents a daily observation in a given city ($n$ = 292); lines represent predicted means. Plots use partial residuals of the response variable and thus show the effect of a given predictor while controlling the effects of remaining predictors.

## DISCUSSION

Our results indicate that the number of confirmed COVID-19 cases in Brazil has been higher and has begun to increase earlier in cities receiving more flights, consistent with the expected role of air-transport connections in spreading the virus across the country (*Ribeiro et al., 2020*). Further, there have been more cases in cities with higher population density, consistent with the expected role of host density (*Poole, 2020*). We also have uncovered a temperature response with a lag of 15 days: there have been more confirmed cases in colder cities and days, and confirmed cases have accumulated faster under lower

**Table 2 Results of the generalized linear mixed model (GLMM) testing for interaction effects on daily cumulative counts of confirmed COVID-19 in Brazil ($n$ = 292; $R_c^2$ = 0.98; $R_m^2$ = 0.77).** The model assumed Poisson-distributed errors and log link, and used capital city identity as a random factor to account for autocorrelated errors of time series within cities. Only statistically significant predictors in Table 1 were used, all of which were scaled to zero mean and unit standard deviation. SD indicates predictor standard deviation; numbers in bold represent statistically significant effects ($P < 0.05$). Variables were as follows: time–time elapsed in days; Log flights—log-transformed number of arriving flights in 2020; density—inhabitants by km$^2$; temperature—mean daily temperature (°C) with a 15-day lag.

| Predictor | SD | Coefficient | $z$ | $P$ |
|---|---|---|---|---|
| Intercept | – | 2.13 | – | – |
| Time | 5.71 | 1.45 | 46.55 | **<0.001** |
| Log flights | 1.19 | 0.96 | 4.67 | **<0.001** |
| Density | 3,057.46 | 0.31 | 1.58 | 0.115 |
| Temperature | 3.11 | −0.30 | −8.91 | **<0.001** |
| Time × log flights | – | −0.08 | −2.46 | **0.014** |
| Time × density | – | 0.01 | 0.47 | 0.641 |
| Time × temperature | – | −0.08 | −2.85 | **0.004** |

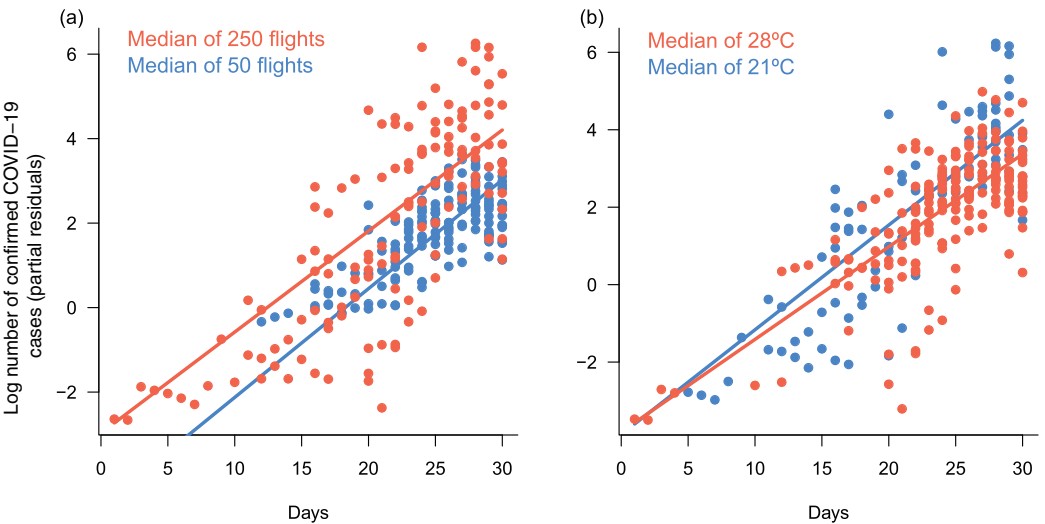

**Figure 5 Response of daily cumulative counts of confirmed COVID-19 cases in Brazil to interactive effects between time an number of arriving flights and mean daily temperature.** Response of daily cumulative counts of confirmed COVID-19 cases across the 27 Brazilian capital cities to interactive effects of (A) time and number of arriving flights, and (B) time and mean daily temperature, based on the Generalized Linear Mixed Model (GLMM) presented in Table 2. The model assumed Poisson-distributed errors and log link, and included capital city identity as a random factor to account for autocorrelated errors in time series within cities. Each point represents a daily observation in a given city ($n$ = 292); lines represent predicted means for each group of observations, as indicated by legends. Group medians were chosen based on their respective predictor ranges (see Fig. 4). Plots use partial residuals of the response variable and thus show the effect of a given interaction while controlling the effects of remaining predictors.

temperature days. Although correlative, these patterns were independent of several demographic and socioeconomic factors and, thus, are unlikely to be confounded by them. This is disturbing because there is little that authorities can do about this relationship,

whereas the number of arriving flights and population density can be manipulated indirectly by isolation strategies.

The temperature dependance of COVID-19 in Brazil agrees with data from China, where warmer weather also seemed to limit the spread of COVID-19 while controlling for population density and per capita GDP (*Wang et al., 2020*). It also agrees with the thermal dependance of viability and transmission demonstrated experimentally for better-studied viruses, such as influenza (*Lowen et al., 2007*) and other betacoronaviruses, for example, SARS-CoV-1 (*Chan et al., 2011*) and MERS-CoV (*Van Doremalen, Bushmaker & Munster, 2013*). While the precise mechanism underlying this pattern requires further study, it may be related to the lipid bilayer of coronaviruses, which becomes increasingly unstable as temperature increases (*Schoeman & Fielding, 2019*).

Recognizing that we are talking about the rate of spread and not necessarily final mortality rates, the information is still important for authorities trying to predict demands on health services. Our best estimate is that a rise of about 1 °C in mean daily temperature reduces the number of COVID-19 cases by about 8%, independently of other factors. Thus, for instance, our results indicate that Cuiabá, with a mean July temperature of about 23.4 °C, and Porto Alegre, with a mean July temperature of 14.9 °C, may differ up to 50% in the number of COVID-19 cases, all else being equal. It also means that the spread in Porto Alegre might be 41% lower in the middle of March (mean daily temperature of 21.3 °C) than it will be in the middle of July (mean daily temperature of 14.9). At the same time, and contrary to some suggestions (*Poole, 2020*; *Wang et al., 2020*; *Grant et al., 2020*), we found no evidence for effects of solar radiation or humidity. Perhaps such conditions are not limiting for the virus or the disease under the climatic conditions of Brazil. Also, rapid evolution of the climatic niche of SAR-Cov-2 could have a similar effect. Although the mutation rate of SARS-CoV seems to be moderate compared to that of other RNA viruses (*Zhao et al., 2004*; *Yang et al., 2020*), clustering of hundreds of worldwide SARS-CoV-2 genomes based on widely shared polymorphisms suggests 16 subtypes, all of which harbor amino acid replacements which may have phenotypic effects (*Morais et al., 2020*).

Whether the temperature effect is related to the rate of spread of SARS-CoV-2 or to the proportion of persons that suffer reportable symptoms cannot be answered with the data being provided at the moment (*Fasina, 2020*). That would require universal testing for the presence of the virus, which is not presently viable but may be a necessity in the following months. Also, one of the main difficulties we encountered was the lack of systematization of current information, since much of the data generated daily is still scattered and difficult to access. For instance, the number of tests performed, which is the key to estimate the rate of infection and the number of infected patients is not available. The large number of publications on the subject in the last 30 days shows that the scientific community is prepared for a quick response, as long as there is a systematization and transparency of information regarding the number of tests being performed, number of suspected cases, number of infected, number of deaths, etc.

Our models are necessarily simple and have limitations. Most importantly, we need city and state administrations to provide the number of performed tests on a regular basis,
so that this variable can be explicitly accounted for in the model. The data do not allow us to investigate complex nonlinear effects, which likely would require data on temperatures beyond those observed in Brazilian cities in March. Also, it is currently not possible to account for potential interactions between COVID-19 and other diseases, particularly influenza, which is seasonal. It may not be necessary to worry about this in the northern hemisphere because the peaks in COVID-19 will occur after the peaks in seasonal influenza. However, the predicted peaks in COVID-19 in the southern hemisphere will occur concomitantly with peaks in seasonal influenza (*Nelson et al., 2006*; *Viboud, Alonso & Simonsen, 2006*). The effects may just be additive, but it is not known whether the simultaneous infection will increase the severity of COVID-19 and therefore the demand for ICUs, and this interaction also could be temperature sensitive. Further, air pollution is known to increase susceptibility to viral respiratory infections (*Ciencewicki & Jaspers, 2007*), but the extent to which it affects the prevalence of COVID-19 is unclear. As COVID-19 consolidates in different cities, it will be possible to reduce uncertainties in relation to the role of temperature and other factors. Nonetheless, our models still performed well as judged by their predictive power, even when ignoring interactions between predictors.

We stress that the temperature effect does not mean that the northern, warmer regions of Brazil should expect fewer complications in their health care system, because such regions also have poorer socioeconomic and sanitary conditions (*Barreto et al., 2011*), and ICUs are concentrated in southern regions (*Rhodes & Moreno, 2012*). Although we found no evidence for an effect of income on the number of confirmed COVID-19 cases, this variable is related to the capacity of cities to respond to the pandemic. Furthermore, apart from elapsed time, the predictors with the largest standardized coefficients were the number of arriving flights and population density (Table 1). Indeed, Manaus, the largest city in Northern Brazil, was the first Brazilian city to declare the collapse of the health system early in April 2020, which is consistent with its large number of arriving flights and large population density but relatively low number of ICUs (*Rhodes & Moreno, 2012*). By contrast, although southern, colder regions have a higher density of ICUs, their situation could be aggravated if social isolation measures are not effectively adopted before and maintained throughout winter in those regions (from June to September). This should be especially important for "favelas", that is, poorer, highly populated neighborhoods with deficient infrastructure, which are presumably at high risk of infection. Thus, we do not present our results as an indication of how hospital demand should be calculated, but as a warning that models for Brazil need to take into account predicted temperatures.

## CONCLUSIONS

Declared as a pandemic by the World Health Organization (WHO), the COVID-19 disease has changed human behavior and strongly affected health systems and the economy worldwide. In an extremely demanding scenario, optimizing the distribution of resources is an essential task. Brazil and other countries are starting to discuss the flexibilization of social distancing policies, as the latter could have important economic costs. However, we need to understand how and when to implement such decisions in

order to prevent new, uncontrolled disease outbreaks that may overcrowd the health care system again and generate even higher economic costs in the near future. Our results suggest that, along with arriving flights and population density, temperature should be taken into account to estimate the number of cases of COVID-19, especially with winter approaching in the southern hemisphere.

## ACKNOWLEDGEMENTS

We are grateful to Álvaro Justen and his collaborators for compiling the daily records on COVID-19 cases for Brazilian cities, and Sara Mortara, Andrea Sánchez-Tapia, Karlo Martins and Gustavo Paterno for developing R packages to facilitate obtention of COVID-19 data from the Brazilian Ministry of Health. We thank the Biodiversity Research Program (*PPBio Brasil*) of the Ministry of Science, Technology, Innovation and Communication (MCTIC) of Brazil for providing the contact network that enabled quick collaboration among the researchers involved in this manuscript. We also thank Alexandre Almeida, Daniel Pimenta and Lucas Bandeira for useful suggestions on preliminary analyses, and Elizabeth Franklin, Daniela Bôlla, Fabíola Wieckert, Sérvio Ribeiro and two anonymous reviewers for useful comments on earlier versions of this manuscript.

### Funding

This work was supported by Ministry of Science, Technology, Innovation and Communication (MCTIC) of Brazil, the Brazilian Agency of Higher Education (CAPES) and the Brazilian National Council for Scientific and Technological Development (CNPq). Pedro Pequeno received a postdoctoral fellowship from the Brazilian Agency of Higher Education (CAPES). Clarissa Rosa and Jorge Luiz Souza received a postdoctoral fellowship from the Institutional Training Program of the Brazilian National Council for Scientific and Technological Development (PCI-CNPq). William Magnusson received a productivity grant from CNPq. The funders had no role in study design, data collection and analysis, decision to publish, or preparation of the manuscript.

### Grant Disclosures

The following grant information was disclosed by the authors:
Ministry of Science, Technology, Innovation and Communication (MCTIC).
Brazilian Agency of Higher Education (CAPES).
Brazilian National Council for Scientific and Technological Development (CNPq).
Brazilian Agency of Higher Education (CAPES).
Brazilian National Council for Scientific and Technological Development (PCI-CNPq).
CNPq.

### Competing Interests

The authors declare that they have no competing interests.

## Author Contributions

- Pedro Pequeno conceived and designed the experiments, performed the experiments, analyzed the data, prepared figures and/or tables, authored or reviewed drafts of the paper, and approved the final draft.
- Bruna Mendel conceived and designed the experiments, performed the experiments, authored or reviewed drafts of the paper, and approved the final draft.
- Clarissa Rosa conceived and designed the experiments, authored or reviewed drafts of the paper, and approved the final draft.
- Mariane Bosholn conceived and designed the experiments, authored or reviewed drafts of the paper, and approved the final draft.
- Jorge Luiz Souza conceived and designed the experiments, authored or reviewed drafts of the paper, and approved the final draft.
- Fabricio Baccaro conceived and designed the experiments, authored or reviewed drafts of the paper, and approved the final draft.
- Reinaldo Barbosa conceived and designed the experiments, authored or reviewed drafts of the paper, and approved the final draft.
- William Magnusson conceived and designed the experiments, authored or reviewed drafts of the paper, and approved the final draft.

## Data Availability

The raw measurements are available in the Supplemental Files.

## Supplemental Information

Supplemental information for this article can be found online at http://dx.doi.org/10.7717/peerj.9322#supplemental-information.

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
