# Peer review of "Air transportation, population density and temperature predict the spread of COVID-19 in Brazil"

_PeerJ, doi:10.7717/peerj.9322_

## Round 0.1 · original submission · Major Revisions

Please revise this manuscript as suggested by reviewers.

·

Basic reporting

This is an interesting article, testing various habitat-related traits of Brazilian cities, from population parameters to socio-environmentally driven effects, and verified the confounding of these, more classic epidemiological traits, with temperature. Air flight connections were also verified as source of invading infected people.
They assumed the disease is new and make assumptions about its delay in spreading in hot temperatures. However, a few of the statements are contradictory to the fact that the virus biology and its fast population dynamic are poorly understood. For instance, they stated the effect of humidity and temperature, likewise lack of vitamin D (thus, few sun exposition), and even the fact that sun deactivates the virus, as important factors. All this may have some relevance to define or circumscribe what makes human habitats susceptible. However, I wish to see more on that concerning uncertainty and how this could be relevant at this moment or for this particular paper. Eventually, a few words on air pollution and sanitarian precarity as well.
An interesting point about this manuscript is that the hottest cities, thus the least vulnerable in Brazil (concerning only this factor), are also the poorest, and worse watched by sanitarian vigilance, about food, hygiene and pollution control in general. Hence, the advantage of having a hot climate, if it is not prohibitive to the disease, may be overcome by the vulnerability caused by these confounding habitat conditions. Authors stated this well by the end of the introduction, but did not rescued these aspects in their discussion, which may be of great interest for decision makers.

Experimental design

- As they modelled from the very first days, indeed they will have the chance to confirm the trends found with time, considering the speed of dissemination of the disease. Also, several non-capital (what were excluded here) are already contaminated, and could provide a further step in analysing the urban environmental parameters in stronger contrast, mostly due to population size variance in a same region. In this sense, the paper defines an excellent starting point for future monitoring.
- Said that, the chosen variables to be test are very good. An excellent choice and a great use of a good public database. Still, a have few questions:
- Why did they use cities as random factors as the independent fixed factors had one measure per city? AS the dependent variable is case counts per time per city (also one measure per time per city), I see the city as the actual replicate level and counts along time as repeated measures. I just wish they would comment and justify the model choice in this concern, or try it in this way I am proposing, if they agree this is a better format. It sounds for me, in terms of structuring the d.f., won´t change mush, however, the most theoretically correct design should be chosen. Apart from this, the analytical design was competently built.
- Why did they not interact significant factors? I see risks of a few important confounding factors, some they pointed in the introduction, that must have been tackled by terms interactions. I will come back to this below.

Validity of the findings

- I have just one concern about their findings and, therefore, conclusions about temperature. The figure 1 shows clearly that the warmer capitals are those in the Amazon, West, and few in Northeast Brazil. Those are the cities with less flight numbers too. Also, they mentioned Ribeiro et al, in preprint, and those authors showed a likely 40 days delay between the South-East capitals getting highly infected and those here showed as the hottest ones. Should they not had interact these terms and tried to conclude whether a temperature effect is indeed happening, or if there is a dynamic itself hiding an unfortunate confounding? In the statement “Nevertheless, a change in 1 ºC predicted 162 a decrease in the number of confirmed cases by [exp(-0.25/3.11)-1 ]×100 = 7.9%, 163 independently of other factors (Table 1).”, they assumed to interpret the temperature independently of other factors, and I don´t see this way.

Still, the predictions made in comparison with extremely different cities for temperature (lines 198-203) is of great importance, but I just wish to see this adjusted to the time expected for the arrival of the COVID-19 in each one of them, that is dependent on flight numbers. A correction here may help authorities to plan even better priorities along time.

Additional comments

- I consider extremely important to fit this confounding or re-calculate whether an actual temperature effect is expected to happen, by correcting its relative importance in contrast with the other significant effects. If at least this is discussed more openly it would help better policy makers to act. My fear is the results as now presented, may pass the false idea that Amazon, for instance, is a less worrying region, and facts are showing otherwise already.

Reviewer 2 ·

Basic reporting

Overall, the basic reporting is ok, with a few editorial corrections noted on review.

Experimental design

As noted, most of the experimental design is ok, but with some concerns regarding testing results and selection of temperature data.

Validity of the findings

While the data is useful for the researchers working on viral disease outbreaks, my concern is that the conclusion that temperature correlated to infection may be skewed by accurate testing as well as overall temperature data.

Additional comments

I have outlines some concerns in the attached review and hope that the suggestions can contribute to an improved manuscript. Specifically, I think that the results section should be expanded with more explanation of the dataset and implications for policy makers.

Annotated reviews are not available for download in order to protect the identity of reviewers who chose to remain anonymous.

Reviewer 3 ·

Basic reporting

Need to describe by more figure
Please use high resolution figures.
Very well written, need to give more attention in unambiguous text.
Explain more result section.

Experimental design

This is an interesting study and the authors have collected a unique dataset using cutting
edge methodology. The paper is generally well written and structured. However, in my opinion the paper has some shortcomings in regards to some data analyses and text, and I feel this unique dataset has not been utilised to its full extent

Validity of the findings

Result should be more elaborative.

---

## Round 0.2 · accepted · Accept

We have finally accepted this COVID-19 article.

Reviewer 2 ·

Basic reporting

The authors have addressed my concerns in the revised manuscript.

Experimental design

The author's considered and implemented additional comparisons and included these data in the revision.

Validity of the findings

The authors explain the most significant results from the data and have greatly improved the discussion section.

Additional comments

The revised manuscript has addressed reviewer comments and I now recommend publication.

Reviewer 3 ·

Basic reporting

na

Experimental design

na

Validity of the findings

na

Additional comments

Acceptable